# A Pharmacological Analysis of the Activity and Failure of the Medical Treatment of High-Grade Osteosarcoma

**DOI:** 10.3390/medicina57020141

**Published:** 2021-02-05

**Authors:** Alessandro Comandone, Antonella Boglione, Tiziana Comandone, Fausto Petrelli

**Affiliations:** 1Department of Medical Oncology, ASL Città di Torino, 10128 Torino, Italy; antonella.boglione@aslcittaditorino.it; 2Italian Group of Rare Tumors, Corso Galileo Ferraris 54, 10129 Torino, Italy; tiziana.comandone.tc@gmail.com (T.C.); faustopetrelli@gmail.com (F.P.); 3School of Specialization of Hospital Pharmacy, University of Torino, 10125 Torino, Italy; 4Department of Medical Oncology, Ospedale di Treviglio, 24047 Treviglio, Italy

**Keywords:** osteosarcoma, chemotherapy, target therapy, pharmacology, resistance

## Abstract

Osteosarcomas (OSs) are a group of neoplasms originating from bone cells, usually presenting in three specific age groups: children, young adults, and the elderly. High-grade OS is an extremely malignant tumor mainly due to evolution into metastatic disease, usually in the lungs. Survival of these patients has improved since the 1980s thanks to close cooperation between oncologists, oncological surgeons and orthopedic surgeons. Unfortunately, no progress has been made in the last 30 years and new, more effective drugs are needed. This article reviews the biological and pharmacological basis of the treatment of OS. Models of clinical pharmacology of the active drugs, toxic effects and reasons for primary and secondary resistance to old and new drugs are discussed.

## 1. Introduction

Osteosarcoma (OS) is the most common primary bone sarcoma in the context of rare tumors. It represents 0.2% of malignant tumors with about 1000 cases per year in the USA [1], and it predominantly affects adolescents, young adults and elderly [2]. Seventy-five percent of the patients are between eight and twenty-five years of age.

OS arises from mesenchymal cells of the bone with osteoblastic phenotype and osteoid production. The tumor is more often localized in long bones [2].

High-grade OS is an extremely malignant tumor: following the surgical treatment alone the tumor is responsible for the death of the patient in about 70% of cases, mainly because of the presence of micrometastatic disease, even in seemingly localized cases [3].

The multidisciplinary approach and the association between chemotherapy (pre- and postoperative) and surgery, increases the event free survival at five years to 60–70% [3].

Neoadjuvant and adjuvant chemotherapy presents several advantages: elimination of micrometastases, better control of the primary tumor, assessment of the effect of chemotherapy, improvement of the prognosis.

Despite these results 30–40% of patients with localized OS treated with chemotherapy and surgery will develop local or distant recurrence during the five years follow up [4]. Eighty per cent of metastases are located in the lungs, local relapse ranges from 7% to 25%, and bone metastases are detected in 10% of cases. Five-year survival for locally recurrent disease ranges from 28% to 33%. On the contrary distant metastatic disease has a very poor survival rate: 10% at 5 years [5].

Resection of lung metastasis can improve long term survival (20% at five years); nonetheless, radical resection can only be carried out in 40% of cases [2,6].

Consequently, when the disease becomes metastatic and non-operable, the only medical treatment that can be provided is palliative in most cases.

Very few active agents are available in metastatic OS and this is the main reason of the disappointing results.

The aim of this paper is to analyze under a pharmacological point of view the mechanism of action of the drugs utilized as first and second line therapy in OS, focusing on the pharmacodynamic and pharmacokinetic aspects of the molecules.

Special attention is paid to the reasons for the onset of resistance to chemotherapy in metastatic disease.

## 2. Chemotherapy

Since the 1980s, few drugs have demonstrated effective activity against OS cells. The most active regimen in the neoadjuvant/adjuvant setting is MAP, a combination of High Dose Methotrexate (HDMTX), Adriamycin (ADM) and Cisplatinum (DDP).

Ifosfamide (Ifo) was integrated into the protocols over the last 20 years [7] as the fourth active drug in postoperative setting in case of poor pathological response to MAP.

A single agent regimen is less effective than combination therapy and many studies have utilized the MAP combination in the neoadjuvant/adjuvant setting as well as in non-pre-treated metastatic disease [2,6].

## 3. Antimetabolites

### 3.1. High-Dose Methotrexate (HDMTX)

Methotrexate is an antifolate inhibiting Dihydrofolate Reductase (DHFR), a key enzyme in folate metabolism. Dihydrofolate reductase is an enzyme that reduces dihydrofolic acid to tetrahydrofolic acid, using NADPH as an electron donor, and it cooperates to maintain the folate pools in purine synthesis.

In humans, the DHFR enzyme is encoded by the *DHFR* gene [8].

The pharmacological effect of MTX is exerted when the drug is polyglutamated. In this intracellular form the antitumor activity is prolonged thereby inhibiting DHFR and thymidylate synthase. MTX can be administered orally or parenterally, however, since absorption in the gastrointestinal tract is highly variable, in oncology MTX is administered intravenously [8,9].

Standard doses range between 40 and 1500 mg/sqm. However, in an effort to improve MTX antitumor activity, doses of HDMTX ranging from 8000 to 15,000 mg/sqm are used in neoadjuvant/adjuvant therapy in OS in order to increase the concentration of the drug in the cells with passive inflow [9].

HDMTX was first administered to OS patients by Jaffe and Rosen in 1970. In 1982, Rosen reviewed the existing postoperative chemotherapy strategies in patients with OS [10]. The same doses of MTX were adopted by COSS *(Cooperative German-Austrian-Swiss Osteosarcoma Study Group)* [11] and by the Rizzoli Institute in Bologna [12]. HDMTX cannot be administered to people over 40 years of age since the reduced liver metabolism and renal excretion would result in high-grade toxicity including mucositis, diarrhea and renal failure, even with leucovorin rescue [13].

### 3.2. Mechanism of Resistance

The mechanism of resistance to MTX can explain the scarce activity of this drug when rechallenged as second line chemotherapy in metastatic OS. The factors involved in resistance include decreased accumulation due to impaired transport, decreased retention as a consequence of lack of polyglutamate formation, increased DHFR, altered DHFR that binds MTX less avidly, and an increased level of a lysosomal enzyme, γ-glutamil hydrolase, that hydrolyses MTX polyglutamates [8,14].

On the contrary, after prolonged exposure to MTX a positive feedback effect can increase the level of DHFR and thymidylate synthase proteins in the cells. This translational effect represents a clinically relevant mechanism following repeated exposure with HDMTX and can play a positive role in case of drug rechallenge [8,14].

## 4. Other Antimetabolites

### Pemetrexed

Pemetrexed is a multitargeted antifolate with a wider range of action than MTX. Its activity involves the inhibition of thymidylate synthase-dihydrofolate reductase (TS-DHFR) and glycinamide ribonucleotide formyltransferase (GRFT) [15,16]. An international study showed very low activity of pemetrexed 500 mg/sqm in 32 patients as second line therapy, with one partial remission and five (15.6%) stable disease. Median progression-free survival (PFS) was 1.4 months and overall survival was 5.5 months and was thus not considered useful in advanced/relapsed osteosarcoma [17].

No other antifolate antipyrimidine agents, including 5-FU, Capecitabine, Cytarabine, 6 Thiopurine, are active against osteosarcoma cells, while marginal activity in second line treatment has been reported for Gemcitabine (see text).

## 5. Antitumor Antibiotics

### 5.1. Doxorubicin

*Doxorubicin*, like all antitumor antibiotics, is a natural product of Streptomyces species with some chemical substitutions that yield a great array of similar compounds. The mechanism of action is broad and includes intercalation into the base pairs of DNA, production of toxic oxygen free radicals with DNA breaks, inhibition of RNA and protein synthesis, defective mitoses, and high mutation rates [18,19].

In the neoadjuvant/adjuvant setting Doxorubicin is actively combined with MTX and Platinum at doses of 45 mg/sqm in a bolus or in a 4-h infusion to reduce cardiotoxicity [7]. As a single agent it achieves no more than 10% of objective response. The main Doxorubicin toxicities include myelosuppression, mucositis, nausea, vomiting, and alopecia. However, the one type of toxicity that has received a great deal of attention is cardiac heart failure or congestive cardiomyopathy. Acute arrhythmia is the most common cardiac toxic effect. Cardiac dilatation, biventricular heart failure and irreversible damage and loss of the primary myocytes may present as late toxicities. The damage is cumulative and cardiac failure shows a sharp and logarithmic increase with the administration of doses of 500–550 mg/sqm. The dose cannot be rescued and the drug has to be discontinued definitively. Considering that Doxorubicin in neoadjuvant/adjuvant settings is administered for at least 9 cycles, the cumulative dose when OS relapses is very close to 500 mq/sqm and further administration cannot be proposed [18,19].

Several options have been put forth in order to increase the maximum tolerable dose: slow infusion of the drug over 4–24 h; in combination with dexrazoxane as an antioxidant agent; introduction of possibly less toxic analogs (epirubicin, lysosomal doxorubicin). Unfortunately, none of these analogs showed increased activity against OS [19].

Moreover, we must also consider the rapid onset of resistance to Doxorubicin. The presence and genetics of the membrane glycoprotein GP170 product expression of ABC multidrug resistance genes that acts as a drug carrier to mediate efflux from neoplastic cells was first described in 1990 [20,21].

Other patterns of resistance to Doxorubicin include reduced binding affinity to topoisomerase II, an increase in the glutathione intracellular pool, and faster detoxification of oxidative species [20,22].

All these reasons prevent rechallenge of metastatic OS with anthracyclines.

No other antitumor antibiotic agents are active in OS cells.

### 5.2. Platinum Analogs (PA)

PA represent a unique class of antineoplastic agents.

Cisplatinum (DDP) is the first approved agent of this group and the only active agent in OS. Platinum has a 2+ or 4+ oxidation state, the ligands around the platinum atom assume a specific geometry. Platinum compounds form strong covalent bonds in the form of an interstrand cross link of DNA, in general between the guanine and adenine bases. DNA replication can be inhibited by platinum adducts [23].

Cisplatinum as a single agent at dose ranging from 80 to 100 mg/sqm every 21 days shows moderate activity on OS cells and in clinical trials, with an objective response no higher than 10%.

In MAP polichemotherapy in the neoadjuvant/adjuvant setting, cisplatinum is fundamental in order to achieve a 65% objective response of the combination [7].

Nephrotoxicity is dose-limiting for cisplatinum resulting in glomerular and tubular damage and magnesium and potassium wasting. Nephrotoxicity is cumulative and can be worsened by the association with acid drugs [23].

Neurotoxicity is another cumulative side effect. Paresthesias, loss of sensitivity in the extremities, pallesthesias and loss of motor function are the most common and irreversible toxicities [23].

Ototoxicity is irreversible as well due to the sensitivity of cochlear hair cells to cisplatinum. No antidote is available for neurotoxic damage [23].

Nausea and vomiting grades 3–4, alopecia and anemia are other very common side effects.

The cumulative dose that is needed to induce irreversible toxicity is about 700 mg/sqm.

Carboplatin, a dicarboxylate platinum, is less neuro- and nephrotoxic but more myelotoxic. Unfortunately, like oxaliplatin, it is inactive against OS cells.

Cisplatinum induces rapid onset of resistance based on reduced cellular accumulation, intercellular detoxification through glutathione synthase, rapid DNA repair via ERCC1 gene expression, and the activation of autophagy [23].

Antineoplastic activity can be determined by microenviroment conditions, hypoxia and DNA mutations.

The rapid onset of resistance, together with irreversible toxicity make cisplatinum unfeasible as a second line agent in metastatic OS.

## 6. Alkylating Agents

Alkylating agents contain reactive alkyl groups that form covalent bonds to DNA, and they include many groups of drugs. The mustard subfamily, which is active in OS, reacts with different atoms of DNA: O_2_; N, S, thereby creating irreversible damage to the double strand.

Of the nitrogen mustard group, cyclophosphamide and more specifically Ifosfamide are the only agents that are actively used. Both drugs are bifunctional agents and interfere with DNA and RNA throughout the whole cellular cycle, though mainly in the S phase [24].

Both cyclophosphamide and Ifosfamide are pro-drugs with no antitumor activity prior to intracellular transformation into amide nitrogen mustard and acrolein (the latter as a toxic metabolite) [24].

In the first studies that were carried out on OS in the 1970′s, cyclophosphamide was the only available agent, and a less than 5% response rate in monotherapy was reported at doses of 600 mg/sqm. Ifosfamide was introduced into the OS regimen when its natural antidote (uromitexan- mesna, a thiol donor) was discovered [6,7,25].

The Italian/Scandinavian Sarcoma Group tried to increase the response rate in the neoadjuvant/adjuvant setting by introducing etoposide and ifosfamide into the perioperative regimen but the results were disappointing: more than 40% of patients developed local or metastatic disease [25].

Currently, Ifosfamide in OS is added to MAP in the postoperative setting in order to improve the clinical outcome when poor tumor necrosis is the result of neoadjuvant therapy [6,25].

A great variety of Ifosfamide doses are considered active: from 1200 mg/sqm over 5 days, to 3 g/sqm/day for three days, up to 15 g/sqm over several days in continuous infusion as second line treatment in metastatic disease.

Ifosfamide toxicity is high (neutropenia, myelotoxicity, renal and bladder toxic effects), but unlike cisplatinum and adriamycin, there is no maximum dose, and the drug can be rechallenged in case of disease relapse [24]. Patients who receive a cumulative dose of 60 g/sqm are prone to have a higher level of toxic effects.

The mechanism of resistance can be partially overcome by increasing the dose of the drug. The reasons for incoming resistance are:changes in drug uptake or transport,increased DNA damage repair,decreased prodrug activation activity,increased scavenging of drug species,increased enzymatic detoxification,altered apoptosis mechanism [24,25,26].

In literature there are some encouraging reports regarding high dose ifosfamide as second line chemotherapy in metastatic OS.

Verschoor et al. demonstrated that second line ifosfamide at a dose ranging from 5 g/sqm to 9 g/sqm resulted in an overall survival of 10.9 months and in 13 months of PFS in the 9 g/sqm arm [26].

At the 2015 ASCO meeting the Italian Sarcoma Group presented a study on high dose ifosfamide as second line therapy in relapsed metastatic OS. Fifty-one heavily pre-treated patients (21 children, 30 adults) were administered Ifosfamide at doses of 15 mg/sqm over 5 days, and in some cases for up to 21 days, with the addition of G-CSF support in all cases. Eleven patients had partial response and 28 achieved stable disease, 6-month progression free survival was 53% and 1- and 2-year overall survival rates were 60% and 31%, respectively [27].

Thus, high dose ifosfamide is now an approved drug for relapsing or metastatic, non-operable OS.

No other alkylating agent has shown any activity in this disease.

## 7. Second Line Chemotherapy Agents

When metastatic OS progresses few other cytotoxic agents are available. Rechallenge, as stated above, is not feasible for most drugs, and other active chemotherapy agents should be used.

The combination of Gemcitabine and Docetaxel is the only polychemotherapy approved as second line treatment.

Gemcitabine is an anti-purine agent that blocks fluorination of the nucleoside and its conversion into an active di-and triphosphate drug. Gemcitabine acts as a fraudulent metabolite reacting as gem-triphosphate with DNA [8]. The drug is metabolized intracellularly by nucleoside kinases into active metabolites: gemcitabine diphosphate and triphosphate, of which the former inhibits ribonucleotide reductase that is implicated in the synthesis of deoxynucleotide triphosphates.

Gemcitabine triphosphates compete as fraudulent antimetabolic agents with deoxycytidine triphosphate for incorporation into the DNA. The final result is the inhibition of DNA synthesis. Gemcitabine has a long persistence into the cell and DNA polymerase cannot remove the drug and repair DNA [8].

Neutropenia, thrombocytopenia, and renal and hepatic toxicity have to be taken into consideration [8].

Gemcitabine resistance can be either primary or acquired. Resistance can result from molecular and cellular changes, including nucleotide metabolism enzymes, inactivation of the apoptosis pathway, high expression of MDR enzymes, activation of the cancer stem cells or enhanced epithelial-to-mesenchymal transition (EMT) pathway [28].

Docetaxel inhibits both free microtubules and spindle separation, arresting cell mitosis in the G and M phases [29]. The pharmacological spectrum after 3 weeks of Docetaxel administration shows tricompartmental pharmacokinetic behavior [29]. Neutropenia, hypersensitivity, fluid retention, and neuromuscular toxicities are the most common side effects. Docetaxel as single agent at 75 mg/sqm every 21 days shows a modest activity with 6% of objective response.

Gemcitabine as single drug at doses of 1250 mg/sqm on day 1 and 8 in OS shows modest activity (<5% response rate), but in combination with docetaxel can determine a 23% rate of clinical control (complete remission + partial remission + stable disease) [30] and a median remission of 8–10 months [31].

As a consequence, gemcitabine-docetaxel combination is a well-recognized and approved second line chemotherapy. The patients received gemcitabine 900 mg/m^2^ on days 1 and 8, and docetaxel 70 mg/m^2^ on Day 8 in 3-week cycles until disease progression or other evidence of treatment failure.

In experimental studies Gemcitabine was utilized in aerosol form in the treatment of lung metastases of OS, assuming that there is direct activity of the drug without metabolism, however, no definitive studies were carried out [32].

No other combinations are considered to be active in second line therapy. Ifosfamide + etoposide, carboplatin + etoposide ± ifosfamide, and cyclophosphamide + topotecan were reported in small phase II studies but activity was not confirmed in larger studies [33,34,35].

High-dose chemotherapy and stem cell support was abandoned after a few studies due to inactivity [36].

Unfortunately, none of the more recent cytotoxic drugs has demonstrated any activity on relapsing or pre-treated OS.

Almost 100 children with pre-treated high-grade OS were included in a data set evaluating new generation drugs: Irinotecan, Topotecan, Imatinib, Ixabepilone and Rebeccamycin analog, but none of these agents were considered active according to conventional response criteria [37].

Trabectedin, a marine derived cytotoxic drug is active in the L form of soft tissue sarcomas, and has been approved as second line therapy. On the contrary, at the same doses of 1.2–1.5 mg/sqm/24 h c-i. q 21 days, no activity was seen in pre-treated OS (no objective response) and the drug was not approved for this indication [38].

Liposome encapsulated muramyl tripeptide-phosphatidylethanolamine (L-MTP-PE) is a liposomal encapsulated analog of muramyl dipeptide that activates macrophages and monocytes, stimulating the immune system in a specific manner.Mori reported a reduced incidence of lung metastases following surgery and an increased survival rate with. L-MTP-PE administered at 2 mg/m^2^ i.v. twice or once weekly as compared to patients treated with CT alone. No further studies were published and the study in the neoadjuvant setting is still ongoing [39].

Zoledronic acid (4 mg q 28 days) is a biphosphonate that inhibits osteoclastic bone resorption and is widely used to reduce cancer-therapy-induced bone loss and osteoporosis in metastatic breast cancer. In OS cell lines, the combination of zoledronate and ifosfamide seems to be synergistic. The results are not encouraging and zoledronic acid in metastatic OS is not recommended [40].

## 8. New Targets and New Agents

While chemotherapy agents destroy cells inhibiting the replication of the tumor, recent advances in our knowledge of biochemical and biological pathways in OS as well as in many other tumors, have led to the recognition of novel mechanisms that could be potentially targeted with molecular drugs.

Several possible OS targets have been identified, but unfortunately none of them are exclusive or easily druggable [41,42].

As a matter of fact, OS displays a great number of genetic, epigenetic and cellular pathway abnormalities with a high degree of intratumor heterogeneity [41,42].

Unfortunately, a widespread, stable genetic lesion among OS types has not been identified.

The most common example of genetic alterations are changes in aploidy and in copy number of genes [42].

Some genetic syndromes are well recognized: Li Fraumeni, Bloom, Werner but none have a specific target to be hit [42].

Somatic syndromes are also represented, but the rarity of these events cannot guarantee they will be a useful target for these therapies.

RB and p53 tumor suppressor genes are in most cases modified and some oncogenes are amplified in high-grade OS: MET, MDM2 (2–25% of cases) cMyc, MPK, PMP22, VEGF A, but none of them can be considered stable druggable targets [41,42].

Epigenetic changes are common with hypermethylation as is histone modification in OS cells, but presently no specific drugs are available [41,42].

As a consequence, targeting some intracellular pathways has become the principal area of investigation for new therapies in OS.

Neoangiogenesis has been the main target of studies over the last ten years. The process of new blood vessel development is critical in tumor growth, normal tissue invasion and metastatic diffusion.

There is a solid rationale for angiogenesis inhibition in OS: both the extent of disease and patient prognosis are correlated with VEGF and VEGFR expression.

In response to hypoxia, tumor tissue releases angiogenetic growth factors such as VEGF, FGF alfa and beta, and PDECG 67 [43].

The primary objective of antiangiogenic therapy is to prevent new vessel sprout and to inhibit tumor growth, invasion and metastatization. The final result is cell dormancy status and tumor regression [44].

Nowadays, we recognize two classes of antiangiogenic drugs: small molecule tyrosine kinase inhibitors acting against different transmembrane receptors, and monoclonal antibodies which act either against circulating growth factors or the extramembrane part of their receptors [45].

In OS therapy only the former category of drugs have been investigated.

On this basis, various targeted agents were studied:–Sunitinib (anti VEGFR 1, 2 and 3, ckit, FTL, CSF 1, RET) [46];–Bevacizumab (BRAF, ckit, FGFR, FLT-3, VEGFA) [47];–Pazopanib (VEGFR 1, VEGFR3, PDGFR) [48];–Sorafenib and Regorafenib provided some relevant benefits [47,48].

Sorafenib, a multitarget tyrosine kinase against VEGF2, RAF 1, BRAF, ckit, FGFR1, FLT3, was investigated at the dose of 400 mg twice a day by Grignani in an Italian Sarcoma Group study [49]. The results were quite interesting: 35 patients were enrolled and 6-month PFS was 45% (95%CI 28–61%, 17 patients). Median PFS and overall survival were 5 (95%CI 2–7) and 11 (95%CI 8–15) months, respectively. Altogether, three (9%) partial responses (PR), two minor responses (6%), twelve (34%) stable disease (SD) for an overall response rate (ORR) of 14% were observed. Sixteen patients were progression-free after 4 months of therapy for an overall PFS at 4 months of 46%.

Regorafenib, another anti VEGFR 1,2,3, TIE 2, KIT, RET, RAF 1, BRAF, PDGFR and FGFR multikinase agent, was investigated by the French Sarcoma Group in a randomized study comparing Regorafenib at an initial dose of 160 mg taken orally on days 1 to 21 of a 28-day cycle with BSC [50]. The study included 43 patients and median PFS was 16.4 weeks (4 months) for the Regorafenib Group, and 4.1 weeks for the BSC group. Median overall survival was 11.3 months in the Regorafenib and 5.9 months in the BSC arm. Toxicity in the Regorafenib arm was mild.

Unfortunately, the results were not considered positive by the European Medicines Agency (EMA) and the drug was not approved for second line therapy of OS.

The Italian Sarcoma Group evaluated the association between Sorafenib and the mTOR inhibitor Everolimus versus Sorafenib alone [51]. Patients received 800 mg Sorafenib + 5 mg everolimus once a day until disease progression or unacceptable toxicity.

The objective response rate was 10% in the Sorafenib arm and 14% in the combination arm.

Sorafenib + Everolimus resulted in 5-month PFS compared to 4 months for the single agent Sorafenib.

The results were considered unsatisfactory and the combination was not approved for OS treatment.

Another negative study is the one by Schwartz [52] which combined Cituxumumab, an anti IGF-1R agent, and Temsirolimus (mTOR inhibitor). Patients received weekly treatment with cixutumumab (6 mg/kg, intravenous) and temsirolimus (25 mg, intravenous flat dose) in 6-week cycles. Only 6 weeks of PFS were reported with an 11% partial response rate.

The most common signs of toxicity of antiangiogenic agents are fatigue, diarrhea, anorexia, oral changes, hand-foot syndrome, thyroid dysfunction, myelotoxicity, and hypertension [45,46,47,48,49,50].

The pharmacological reasons for tumor resistance to TKI are well known: VEGFR and PDGFR gene mutation, overexpression of targets, impaired membrane transport by the drug into the cell, accelerated drug clearances [41,42,53].

## 9. Check Point Inhibitors

All the studies that have been published to date on this topic are disappointing.

The SARC 028 trial reported only one partial response out of 22 patients who received Pembrolizumab as a single agent in second line therapy in progressing OS. All patients were treated with 200 mg intravenous pembrolizumab every 3 weeks. Median PFS was 24% eight weeks after the start of treatment [54]. Immunotherapy in metastatic, pre-treated high-grade osteosarcoma is not taken into consideration outside clinical studies.

## 10. Conclusions

Osteosarcoma therapy made a great deal of progress in the 1980′s following the discovery of MAP as an active combination of drugs for frontline treatment, both in primary and in metastatic disease.

Ifosfamide came later and added some interesting results in the postoperative setting in poor responder patients.

Perioperative chemotherapy improved the results and changed the prognosis in the majority of patients, increasing 5-year survival from 20% to 65%.

Unfortunately, when the disease relapses, second line therapy provides far from satisfactory results.

The MAP combination as second line is scarcely active because of the onset of secondary resistance, and the established maximum dose of Adriamycin and Platinum which hampers rechallenge.

Few other chemotherapy drugs have shown much activity: Docetaxel and Gemcitabine, alone or in combination, provided an objective response of 23% and a PFS of 4–6 months.

Targeted drugs have a good rationale because of the strong expression of intracellular pathways in OS cells, but the results are partial, erratic and short lasting.

At present, immunotherapy shows no benefits.

Forty years after the seminal study of Jaffe et al., the current standard therapy in OS is the combination of old cytotoxic drugs with surgery of both the primary tumor and of the resectable lung metastases. Integrated therapy has radically changed the prognosis in high-grade OS. Five-year survival is now 70% compared to 20–30% with surgery alone. Radiotherapy plays only a palliative role.

Unfortunately, no new active agents have been identified in the last 30 years. Newer cytotoxic agents, as well as molecular targeted agents have shown no benefits and none of the targeted agents have been approved. Immunotherapy plays no role in OS.

There is an urgent need for effective agents, especially to treat metastatic or relapsing disease.

Basic research, active international cooperation, and a multidisciplinary approach are the key choices for future improvements in therapy.

## Data Availability

Publicly available datasets were analyzed in this study.

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
