# Peer review of "A Pharmacological Analysis of the Activity and Failure of the Medical Treatment of High-Grade Osteosarcoma"

_medicina, 2021, doi:10.3390/medicina57020141_

Round 1

Reviewer 1 Report

Introduction is not well written, is chaotic, not easy readable, some of information could be misleading – osteosarcoma predominantly affects adolescent, young adult and elderly not infants; currently 5-y survival is 70% - 65%, but in the introduction “the tumor is responsible for the death of 70% of cases”. Additionally, the one sentence is repeated. The introduction needs major revision.

Ifosfamide –

  1. maximum cumulative doses are defined; patients who receive cumulative doses of less than 60 g/m2 are at lower risk of toxicity; it should be clarified.
  2. The sentence about ifosfamide and MAP (Alkylating agents Section) should be corrected or/and comments should be added – according the ESMO Guidelines Committee ref.2 „ case of poor pathological response to the preoperative MAP regimen, the postoperative addition of ifosfamide and etoposide to MAP failed to improve the survival and increased the risk of secondary malignancy compared with those patients treated with the MAP regimen only [I, C]

The role of the docetaxel should be clarified – in the same Section is active and inactive in a second line therapy (page 6).

Author Response

I totally agree that Introduction was not easily readable. We introduced the changes  requested  and  re wrote the paragraph. Thank you.

Ifosfamide has not a real top dose , but many Authors report the increase of toxicity above 60 mg/sqm.

Remark accepted.

 Ifosfamide  as a matter of fact  is  the forth most active drug in OS during  perioperative therapy. The Scandinavian and Italian sarcoma group protocol added the drug if the pathological response was less than 90% as a  salvage therapy.  Remark accepted.

Docetaxel  as single agent has a low activity ( 6% objective response) but in combination with Gemcitabine increases the objective response of the polichemotherapy to 23%. Remark accepted. Text changed

Reviewer 2 Report

  1. Short Summary of the drugs that are used for OGS
  2. The ADM is generally converted to the generic of doxorubicin. So would consider changing adriamycin to doxorubicin. Not all pt get adriamycin, but all get doxorubicin.
  3. Dosing schedules are alluded to but not much data on exploring the studies in detail, but give short rational why doses were selected and outcomes were provided.
  4. The new agents address succinctly.

Author Response

The  drugs active in OS are listed in the abstract, in the introduction and in the following part of the article. Revision accepted

In every part of the article the term Adriamycin has been changed in Doxorubicin. Remark accepted

The doses of the drugs have been reported for every drug. Thank you for the remark Accepted

The paragraph on  the  new drugs  is short because: none of them was approved by the regulatory agencies  ( USA & EU) and none of them demonstrated a concrete  activity against OS

Furthermore the potential targets of the drugs are widely presented in the paragraph  “New target and new agents”  explaining the targets,  the molecular action of the  agents, .and the mechanism of resistance

In every drug the optimal dose was reported as the Reviewer requested for the biological agents too.

Thank you for the suggestions.
